# Dimethyl Fumarate Attenuates Lung Inflammation and Oxidative Stress Induced by Chronic Exposure to Diesel Exhaust Particles in Mice

**DOI:** 10.3390/ijms21249658

**Published:** 2020-12-18

**Authors:** Isabella Cattani-Cavalieri, Helber da Maia Valença, João Alfredo Moraes, Lycia Brito-Gitirana, Bruna Romana-Souza, Martina Schmidt, Samuel Santos Valença

**Affiliations:** 1Institute of Biomedical Sciences, Federal University of Rio de Janeiro, Rio de Janeiro 21044-020, Brazil; i.cattani.pinto.cavalieri@rug.nl (I.C.-C.); helfarma@yahoo.com.br (H.d.M.V.); jmoraesbr@yahoo.com (J.A.M.); lyciabg@gmail.com (L.B.-G.); samuelv@icb.ufrj.br (S.S.V.); 2Department of Molecular Pharmacology, University of Groningen, 9700 Groningen, The Netherlands; 3University Medical Center Groningen, Groningen Research Institute for Asthma and COPD (GRIAC), University of Groningen, 9700 Groningen, The Netherlands; 4Department of Histology and Embryology, Rio de Janeiro State University, Rio de Janeiro 20943-000, Brazil; romanabio@gmail.com

**Keywords:** air pollution, diesel exhaust particles, dimethyl fumarate, inflammation, oxidative stress, lung, mice

## Abstract

Air pollution is mainly caused by burning of fossil fuels, such as diesel, and is associated with increased morbidity and mortality due to adverse health effects induced by inflammation and oxidative stress. Dimethyl fumarate (DMF) is a fumaric acid ester and acts as an antioxidant and anti-inflammatory agent. We investigated the potential therapeutic effects of DMF on pulmonary damage caused by chronic exposure to diesel exhaust particles (DEPs). Mice were challenged with DEPs (30 μg per mice) by intranasal instillation for 60 consecutive days. After the first 30 days, the animals were treated daily with 30 mg/kg of DMF by gavage for the remainder of the experimental period. We demonstrated a reduction in total inflammatory cell number in the bronchoalveolar lavage (BAL) of mice subjected to DEP + DMF as compared to those exposed to DEPs alone. Importantly, DMF treatment was able to reduce lung injury caused by DEP exposure. Intracellular total reactive oxygen species (ROS), peroxynitrite (OONO), and nitric oxide (NO) levels were significantly lower in the DEP + DMF than in the DEP group. In addition, DMF treatment reduced the protein expression of kelch-like ECH-associated protein 1 (Keap-1) in lung lysates from DEP-exposed mice, whereas total nuclear factor κB (NF-κB) p65 expression was decreased below baseline in the DEP + DMF group compared to both the control and DEP groups. Lastly, DMF markedly reduced DEP-induced expression of nitrotyrosine, glutathione peroxidase-1/2 (Gpx-1/2), and catalase in mouse lungs. In summary, DMF treatment effectively reduced lung injury, inflammation, and oxidative and nitrosative stress induced by chronic DEP exposure. Consequently, it may lead to new therapies to diminish lung injury caused by air pollutants.

## 1. Introduction

Air pollution is a global problem that can cause several health issues, mainly affecting the respiratory system [1,2,3,4]. One of the main components responsible for air pollution is the combustion of diesel fuel [5]. Exacerbations of chronic respiratory diseases, such as asthma and chronic obstructive pulmonary disease (COPD), involve airway inflammation and oxidative stress, processes that can be triggered by exposure to air pollution [2,3,4,6,7,8,9,10]. Diesel fuel combustion is particularly harmful due to the release of diesel exhaust particles (DEPs) in addition to other highly toxic compounds, such as metals and polycyclic aromatic hydrocarbons. DEPs with a diameter less than 2.5 µm have the ability to penetrate deeply into the lungs, reaching the alveoli [6,11].

Exposure to DEPs has been demonstrated to be an important factor in the induction of inflammatory processes and oxidative stress in mice [12,13]. Indeed, one of the reasons for the progression of chronic respiratory diseases by air pollution—including DEP—is the induction of inflammation [14,15], a process studied in a range of experimental models. For example, exposure of mice to DEP from a light medium duty Euro 1 diesel engine by intranasal instillation elevated macrophages and neutrophils in bronchoalveolar lavage (BAL) measured 6 and 24 h after exposure to DEP [12]. In addition, exposure of mice to DEP (SRM2975) from the National Institutes of Standards (NIST) by intratracheal installation elevated macrophages and neutrophils in BAL and elevated the lipid peroxidation marker malondialdehyde measured 24 h after exposure to DEP [13]. Moreover, chronic exposure of rats to DEP with 3 mg/m^3^ from diesel engine exhaust elevated the pro-inflammatory cytokines interleukin (IL)-8, IL-6, and tumor necrosis factor (TNF)-α in BAL, serum and lung homogenates, and elevated total number of inflammatory cells, neutrophils, eosinophils and lymphocytes [16]. Additionally, mice exposed to diesel exhaust for one month with low (100 mg/m^3^) and high (3 mg/m^3^) showed increased levels of TNF-α IL-4 and IL-10 mRNA in lung tissue [17]. Exposure of mice to DEP by intranasal instillation for 30 days and 60 days, showed the elevation of total inflammatory cell number in BAL, whereas differential cell counts in the BAL showed no differences [18]. In addition, exposure of mice to DEP (SRM 2975; NIST) by intratracheal installation increased the protein level of the nuclear factor kappa B (NF-κB) p65 measured 24 h after exposure to DEP [19]. Moreover, alveolar macrophages exposed to DEP (SRM 2975; NIST) showed elevated protein level of nuclear NF-κB p65 after DEP exposure [20]. Next to the induction of inflammation by DEP, induction of oxidative stress by DEP seems to represent a permissive factor for the progression of obstructive pulmonary disorders [9,10,21]. Oxidative stress can activate nuclear factor erythroid-derived-like 2 (Nrf2) [22,23], an important transcription factor involved in the protection of cells against oxidative stress. Inactive Nrf2 is bound to its suppressor kelch-like ECH-associated protein 1 (Keap-1). Upon activation, Nrf2 is released and induces phase II enzyme and antioxidant gene activation, promoting detoxification of the cell in order to reduce oxidative damage. Exposure of mice to DEP from the vent of a workshop in a diesel engine manufacturing factory by intratracheal instillation increased the expression of both *Nrf2* gene and Nrf2 protein in the lung measured up to 48 h after exposure to DEP [24]. Recently, our group demonstrated that mice exposed to biodiesel particulate matter elevated Nrf2 protein expression in the lung after 5-day exposure to the pollutant [25].

Dimethyl fumarate (DMF) is known as an oral fumaric acid ester with anti-inflammatory and antioxidant properties [26,27]. Recent studies have explored the use of DMF in models for systemic inflammation, lung diseases and neuronal disorders [28,29,30,31,32]. One of the actions of DMF is through modulation of NF-kB signaling pathway. By inhibiting NF-kB translocation and activation, it interferes with the normal functions of NF-kB in inflammatory responses, including the production of proinflammatory cytokines [26,27,33]. In addition to its anti-inflammatory effects, DMF is also involved in oxidative stress reduction by inducing Nrf2-mediated neurocytoprotection [34,35]. In an acetaminophen-induced hepatic injury model in mice, DMF reduced elevated protein levels of NF-κB and Nrf2 in liver homogenates [28]. Despite its anti-inflammatory and antioxidant properties [26,27], DMF has not been studied in experimental models exposed to air pollution—including DEP—yet. The purpose of this study was to evaluate the therapeutic potential of DMF to ameliorate lung injury, inflammation, and oxidative stress in mice chronically exposed to DEP.

## 2. Results

### 2.1. DMF Reduces Total Inflammatory Cell Number in BAL and Lung Injury Caused by DEP Exposure

To evaluate the effects of DMF on inflammation after chronic DEP challenge, the numbers of total inflammatory cells and macrophages were quantified in BAL (Figure 1B) and sections (Figure 2D), respectively. The total inflammatory cell number was increased in the DEP and DEP + DMF groups when compared to controls (Figure 1B). Importantly, DMF treatment significantly reduced the total inflammatory cell number in animals challenged with DEP (Figure 1B). Histological analysis of alveoli revealed that the number of macrophages was higher in the DEP and DEP + DMF mice than in the control group, and that DMF treatment did not significantly affect macrophage infiltration induced by DEP (Figure 2A–D). To assess the effect of DMF on lung injury caused by DEP exposure, we implemented a five-point lung injury scoring system (Figure 2E–H). Lung injury was scored significantly higher in the DEP and DEP + DMF groups when compared to controls (Figure 2H). However, DMF treatment markedly reduced DEP-induced lung injury by ~45% (Figure 2H).

### 2.2. DMF Reduces Oxidative and Nitrosative Stress after DEP Instillation

We next examined whether DMF treatment could affect the oxidative and nitrosative stress induced by DEP challenge. For this purpose, we analyzed total ROS in the BAL of our experimental groups: total ROS in the DEP and DEP + DMF groups was increased when compared to controls, but was significantly lower in the DEP + DMF than in the DEP group (Table 1). Intracellular total ROS, OONO and NO levels were elevated by chronic DEP exposure; importantly, levels were normalized to baseline by DMF treatment (Table 1).

### 2.3. DMF Treatment Decreases DEP-Induced Keap-1 and NF-κB Expression in Mouse Lungs

To evaluate the effects of chronic DEP exposure and DMF treatment on the Nrf2 pathway, we measured Nrf2 and Keap-1 protein expression in whole lung lysates by Western blotting. The expression of Nrf2 was unchanged in all groups (Figure 3A,E). However, DEP challenge increased Keap-1 protein expression, which could be normalized by DMF treatment (Figure 3B,E). Total NF-kB p65 protein expression in the lung was significantly increased by chronic DEP exposure, and reduced to levels below baseline in the DEP + DMF group (Figure 3C,E). The protein expression of total NF-kB p50 and p-NF-Κb p50 were unchanged in all groups (data not shown). The expression of p-NF-Κb p65 was also unchanged in all groups (Figure 3D,E).

### 2.4. DMF Attenuates the Effects of Chronic DEP Exposure on the Expression of Nitrotyrosine and Redox System-Associated Proteins in Mouse Lungs

To investigate the effects of DMF on DEP-mediated nitrosative damage and changes in the redox profile, we measured the lung expression levels of nitrotyrosine and components related to the redox system. DEP exposure increased nitrotyrosine, Gpx-1/2, and catalase expression, which could be effectively inhibited by DMF treatment (Figure 4A–C,E). Surprisingly, SOD-1 protein expression was reduced in the DEP + DMF group when compared to the control and DEP groups, whereas DEP exposure alone had no effect on SOD-1 levels (Figure 4D,E).

## 3. Discussion

In the present study, we aimed to explore the therapeutic potential of DMF against pulmonary damage caused by chronic DEP exposure in mice. Despite its well-documented anti-inflammatory and antioxidant properties [26,27,28,31], DMF has not yet been studied in experimental models of lung injury caused by air pollutants. Therefore, we assessed the influx of total inflammatory cells in BAL and macrophages in alveoli, as well as the lung parenchyma architecture. Importantly, the presence of oxidative and nitrosative stress and redox system markers was determined. Moreover, intranasal instillation is a well-accepted method used for DEP exposure in animal models [36]. We chose to administer DMF by oral gavage in chronically DEP exposed mice to more closely resemble clinical procedures. Additionally, we decided to start the DMF treatment after 30 days of DEP exposure because we aimed to mimic a “real life” situation in which treatment would not be initiated unless there was a probable cause (i.e., previous subchronic exposure to DEP). Thus, we are constantly exposed to air pollution, rendering pretreatment impossible, and typically only seek help after a health exacerbation or problem. Our results demonstrate that DMF has the therapeutic potential to ameliorate lung inflammation and oxidative and nitrosative stress induced by chronic DEP exposure.

Our data showed that chronic DEP exposure resulted in an increased number of total inflammatory cells in BAL. This is in accordance with previous studies, where total inflammatory cell numbers were increased in mice and rats chronically exposed to DEP and diesel engine exhaust of a 4-cylinder diesel engine, respectively [16,20]. We show here that DMF treatment reduces the increase in total inflammatory cell number in BAL induced by DEP, while mice are still being exposed to DEP during treatment, indicating an effect of DMF on inflammatory cell infiltration. An elevation in BAL inflammatory cell number could be considered as an important marker for the inflammatory response of the lung to injury during air pollution exposure. Induction of macrophages by particulate air pollution is related tissue repair and particle phagocytosis [37,38]. Consistent with previous studies [16,39], we observed an increase in macrophages in lung tissue after chronic DEP exposure. Treatment with DMF did not appear to affect macrophage number as shown by histological staining in alveoli. This may indicate that to significantly reduce the number of macrophages, a more prolonged treatment time or higher dose of DMF would be necessary. In addition, we speculate that DMF primarily reduces the induction of oxidative stress by DEP rather than inflammatory cell accumulation into the lungs.

DMF is known to act as an immunomodulator by interfering with NF-kB signaling and is capable of activating the antioxidative Nrf2 pathway [26,34,35]. Nrf2 is bound to and inhibited by Keap-1, which ultimately leads to its ubiquitination and degradation [40]. However, DMF has the ability to bind to Keap-1, releasing Nrf2 and allowing the start of cellular detoxification. For this reason, we decided to evaluate the effects of DEP exposure and DMF treatment on NF-kB, Nrf2 and Keap-1 protein levels. To our surprise, DEP exposure, either with or without DMF treatment, did not affect Nrf2 protein levels. In contrast, our recent study demonstrated an elevation of Nrf2 expression after a 5-day exposure of mice to biodiesel particulate matter, albeit with a much higher dose (250 µg) than used in the present study [25]. Interestingly, we did observe an elevation in Keap-1 protein levels after chronic DEP exposure, which could be reduced to basal levels by DMF treatment (Figure 3). Even though the Nrf2 protein expression was unchanged in all groups, the fact that DMF was able to inhibit the DEP-induced increase in Keap-1 suggests increased availability (or at least similar as under control conditions) of Nrf2. As expected, we found that DEP-induced total NF-kB p65 protein upregulation was suppressed after DMF treatment. Exposure of mice to DEP from a 210 hp engine bus (Mercedes Benz MB1620) by intranasal instillation for 30 days showed increased NF-kB p65 positive cells by immunohistochemistry analysis in alveolar septa of mice [39]. Additionally, augmented p-NF-kB expression was observed after a 5-day exposure to 250 µg of biodiesel particulate matter in mice [25]. Accordingly, mice exposed to DEP (SRM 2975; NIST) by intratracheal installation showed elevated protein levels of NF-kB p65 measured 24 h after DEP exposure [19]. Although we did not observe alterations of p-NF-kB p65 by DEP exposure, our current results suggest a potential role for total NF-kB p65 in the induction of inflammation over a long-term period (60 days) of DEP exposure in mouse lungs. Second, DMF inhibits the DEP-induced increase in total NF-kB p65 expression below baseline. In line with our findings, an acetaminophen-induced hepatic injury model in mice DMF reduced the elevated NF-kB protein level, however, and most importantly, its inhibitory effect was more pronounced in a group exposed twice to DMF [28]. Taken together, these findings implicate that DMF bear the potential to effectively interfere with total NF-kB p65 as key regulator in inflammation which is associated with reduced promotion of the inflammatory response.

An important finding of our work is that exposure of mice to a relatively low dose (30 µg) of DEP [41] (without any additional stimulus) for 60 consecutive days was able to promote lung injury. This is in line with a previous study showing that acute exposure of particulate matter induces lung injury in mice [42]. These are clinically relevant observations, especially considering air pollution may be associated with worsening disease symptoms (exacerbations) in asthma and COPD patients [1,3,4,7]. Despite the lung injury score still being significantly higher in the DEP + DMF group than in controls, DMF treatment was able to markedly reduce DEP-induced lung injury by approximately 45%. This reduction in lung injury may be linked to the antioxidant properties of DMF. As described in the literature, overproduction of ROS and/or reactive nitrogen species (RNS) together with the suppression of antioxidant systems, leads to oxidative and nitrosative stress, which may result in (severe) cellular damage and even cell death [43]. As lung damage by DEP exposure is likely caused by oxidative and nitrosative stress to an important extent, we investigated the impact of DMF treatment on stress markers. In the current study, we showed the ability of chronic DEP exposure to induce both total and intracellular total ROS in BAL cells. Accordingly, a recent study showed that repeated exposure of mice to particulate matter extract induced the production of intracellular ROS [44], and our group demonstrated the ability of acute exposure (5 days) to a mixture of biodiesel and diesel particulate matter in the induction of ROS production in mouse lungs [25]. Here, we show that DMF suppressed total ROS induced by chronic DEP exposure. Strikingly, the increase in intracellular total ROS, as determined by using the H_2_DCFDA probe (considered a more sensitive assay), was fully reduced by DMF to control levels. Similarly, DMF inhibited DEP-induced nitrosative stress, as indicated by a normalization of OONO and NO (Table 1). In addition to these markers of nitrosative stress, nitrotyrosine can be used as an indicator of RNS activity, inflammation and cellular damage [45]. In lung homogenates obtained from the DEP group, we observed the anticipated elevation in nitrotyrosine protein, which could be reduced to basal levels by DMF treatment. Taken together, these results highlight the effectiveness of DMF treatment in reducing oxidative and nitrosative stress caused by chronic DEP exposure. These effects may be associated with DMF inhibiting inflammatory response of the lung to DEP-mediated injury.

The antioxidant system constitutes the first line of defense against oxidants and includes SOD, catalase and Gpx. Both Gpx-1/2 and catalase expression levels were increased in response to DEP exposure, which could be fully prevented by DMF treatment (Figure 4). Interestingly, a recent study in young mice demonstrated no effect of diesel exhaust exposure (30 days) on Gpx activity in lung tissue [46]. Based on our findings, it is conceivable that both GPx-1/2 and catalase were upregulated in defense against the oxidative stress promoted by DEP exposure (evidenced by ROS, ONOO and NO; Table 1). Vice versa, because DMF treatment decreases oxidative stress, we suggest that Gpx-1/2 and catalase protein expression was reduced (or prevented) since their action was not required in the presence of DMF. SOD is one of the most important enzymes to act against ROS, and is considered to be the first enzyme activated in the antioxidant defense system [47]. DEP exposure alone did not change the expression of SOD-1, while DMF treatment of DEP-exposed mice reduced its expression below control levels. Conversely, mice exposed to diesel exhaust for 30 days showed an elevation of SOD activity in lung tissue [46]. It is feasible that because of its role as the first enzyme acting against ROS, SOD-1 expression after 60 days of exposure was already reduced because of the progressed lung response (i.e., antioxidant system moved on to the next step) against the prolonged oxidative stress promoted by DEP. Further studies on the differential impact of DMF on inflammatory responses—including in depth studies on macrophage functions—and on the antioxidant system are certainly an interesting research area. However, such studies are beyond the scope of our current manuscript but obviously are important in follow-up research.

DMF is a well-known drug [26,27]. However, despite its well documented anti-inflammatory and antioxidant properties, DMF has not been studied in experimental models of lung injury caused by air pollutants yet. In our current study, we omitted the inclusion of a DMF group alone. The lack of a DMF group may represent a potential limitation of our current study. However, it is important to note that DMF administered alone to a acetaminophen-induced hepatic injury model in mice showed no toxic effect on liver function measured by biochemical markers such as serum glutamic oxaloacetic transaminase, serum glutamic pyruvic transaminase, gamma-glutamyl transferase, alkaline phosphatase, bilirubin and albumin in serum [28], implicating that DMF alone does not exhibit unwanted side effects. However, and most importantly, we administered DMF by oral gavage in chronically DEP-exposed mice to more closely resemble clinical procedures. Upon addition of DMF after 30 days of DEP exposure, we probably mimic a “real life” situation more closely. We are constantly exposed to air pollution, rendering pretreatment impossible, and typically only seek help after worsening of disease symptoms such as exacerbations of asthma and COPD patients [4,7,8]. It is important to note that DMF reduced lung inflammation and oxidative and nitrosative stress induced by chronic DEP exposure, processes known to be difficult to control and contribute to decline in lung function in asthma and COPD patients. It is also important to note that we used in our current study DEP from NIST, instead of particulate matter from buses [25], to increase the probability of reproducing our current research findings by research groups worldwide. DEP from NIST is a well-studied compound [13,19,20,25]. However, it may not reflect the current fleet in developed countries, indicating a limitation of our study. Specially, increased regulation from environmental agencies to reduce pollutants, as well as the evolution of motor technologies, affects the motor emissions. However, in developing countries, the regulation for motor emissions is not well established or the surveillance is low by environmental agencies parallel to use of old technology [48].

## 4. Material and Methods

### 4.1. Animals

C57BL/6 male mice (8 weeks old; 18–25 g) were obtained from the Laboratory Animal Breeding Center at UFRJ (Rio de Janeiro, Brazil) and housed in a room with a 12 h light/12 h dark cycle. Mice were acclimatized for 2 weeks before the experiments began. The animals had free access to food and water. All animal experiments were approved on 16 January 2019 by the Ethics Committee on Animal Use of the Health Sciences Center of the Federal University of Rio de Janeiro with identification number 01200.001568/2013-87.

### 4.2. Diesel Exhaust Particles (DEPs)

DEPs (SRM 2975) were obtained from the National Institute of Standards and Technology (NIST) (Gaithersburg, MD, USA) and purchased from Sigma-Aldrich (St. Louis, MO. USA) and were suspended in a solution of saline (0.9% NaCl 98% *v*/*v*) and dimethyl sulfoxide (2% *v*/*v*). To minimize aggregation, particle suspensions were sonicated for 15 min and vortexed prior to intranasal instillation. As reported in the Certificate of Analysis from NIST, the mean diameter particle size of DEP (SRM 2975) was 1.62 ± 0.01 μm as analyzed by number distribution.

### 4.3. Dimethyl Fumarate (DMF)

DMF was purchased from Sigma-Aldrich (St. Louis, MO, USA) and prepared in 0.5% carboxymethylcellulose (Sigma-Aldrich). DMF solution and vehicle (0.5% carboxymethylcellulose) were administered daily by oral gavage from day 30 until the end of the experiment.

### 4.4. Experimental Design

Animals were randomly allocated to three groups (8 animals per group): the control group, intranasally instilled with 25 μL of vehicle (98% saline and 2% dimethyl sulfoxide) for 60 consecutive days and treated with vehicle (0.5% carboxymethylcellulose) by gavage from day 30 until the end of the experiment; the DEP group, intranasally instilled with 25 μL of 30 μg of DEP solution for 60 consecutive days and treated with vehicle (0.5% carboxymethylcellulose) by gavage from day 30 until the end of the experiment; the DEP + DMF group, intranasally instilled with 25 μL of 30 μg of DEP solution for 60 consecutive days and treated with 30 mg/kg of DMF by gavage from day 30 until the end of the experiment (Figure 1A). DMF or respective vehicle were given at the morning, and DEP or respective vehicle was instilled after lunch. DMF used here was based on some studies that used doses varying from 10 to 100 mg/kg per animal [28,29,30,31,32]; we chose to treat our animals with a submaximal dose of 30 mg/kg.

### 4.5. Bronchoalveolar Lavage (BAL) and Lung Homogenates

Twenty-four hours after day 60, the animals were exposed to sevoflurane for 2 min (Laboratório Cristália, São Paulo, Brazil) and immediately euthanized by cervical displacement. BAL was performed in the lungs of all animals in each group. The trachea was exposed and cannulated to infuse the right lung with 0.5 mL of buffered saline solution three consecutive times (final volume 1.2–1.5 mL). BAL was collected, centrifuged, and stored on ice. The total number of cells in the BAL was determined using a Neubauer chamber. The right lung of each animal was collected and homogenized in lysis buffer (protease inhibitor cocktail from Sigma + PBS 7.5), and the total protein concentration was determined using the Bradford method. The lung homogenates were used to perform Western blotting.

### 4.6. Reactive Oxygen Species (ROS) and Probe Analysis for Intracellular Total ROS, Peroxynitrite (OONO) and Nitric Oxide (NO)

ROS detection was performed in BAL cells following the adapted nitroblue tetrazolium chloride protocol [49]. BAL cells from 3 randomly chosen samples of each experimental group were used for detection of intracellular total ROS by 2′,7′-dichlorodihydrofluorescein diacetate (H_2_DCFDA), peroxynitrite (OONO) by aminophenyl fluorescein (APF), and nitric oxide (NO) by 4-amino-5-methylamino-2′,7′-difluorofluorescein diacetate (DAF-FM Diacetate) following the manufacturer’s instructions (Thermo Fisher Scientific, Waltham, MA, USA). Briefly, BAL cells and probe samples were incubated for 1 h in a 96-well black plate. After 3 washes in phosphate-buffered saline, the fluorescence reading was performed. Fluorescence was monitored for up to 90 min incubation through excitation and emission wavelengths of 495 and 530 nm, respectively. The 70th minute of the reading was chosen to represent the results on a graph containing the area under curve (AUC).

### 4.7. Tissue Preparation and Microscopic Analyses

The left lungs of each group embedded in paraffin were sectioned (5 μm thick) and the sections were stained with hematoxylin and eosin for histological analyses. Lung injury was scored in ten randomly chosen fields of tissue sections per animal, using a 5-point lung injury scoring system (perivascular and peribronchial inflammation, hyaline membranes, alveolar and interstitial infiltrates, and alveolar hemorrhage) [50]. The number of macrophages were counted in alveoli; ten random fields per animal were analyzed in the lung. Data are expressed as the number of macrophages per mm^2^.

### 4.8. Immunoblotting

Whole lung lysate proteins (30 μg) from 3 randomly chosen samples of each experimental group were separated by sodium dodecylsulfate-polyacrylamide and transferred to PVDF membranes. It is important to note that protein expression of kelch-like ECH-associated protein 1 (Keap-1) and nitrotyrosine represented from 6 randomly chosen samples of each experimental group. After 1 h of blocking in non-fat milk (5%) membranes were probed overnight at 4 °C with the following antibodies: goat polyclonal catalase (64 kDa) (Santa Cruz Biotechnology, Dallas TX, USA; 1:500), rabbit polyclonal glutathione peroxidase-1/2 (Gpx-1/2) (22–31 kDa), goat polyclonal superoxide dismutase 1 (SOD-1) (23 kDa), rabbit polyclonal nitrotyrosine (85 kDa) (Santa Cruz Biotechnology, Dallas TX, USA; 1:500), rabbit polyclonal Nrf2 (61 kDa) (Santa Cruz Biotechnology, Dallas, TX, USA; 1:200), rabbit polyclonal kelch-like ECH-associated protein 1 (Keap-1) (69 kDa) (Santa Cruz Biotechnology, Dallas, TX, USA; 1:500), goat polyclonal NF-κB p65 (65 kDa) (Santa Cruz Biotechnology, Dallas, TX, USA; 1:500), rabbit polyclonal phospho-NF-κB p50 (50 kDa) (Santa Cruz Biotechnology, Dallas, TX, USA; 1:1000), or mouse monoclonal β-actin (42 kDa) (Sigma-Aldrich, St. Louis, MO, USA; 1:1000). After washing, membranes were incubated with the appropriate horseradish peroxidase-conjugated secondary antibodies (Sigma-Aldrich, St. Louis, MO, USA; 1:100). The antigen–antibody complexes were detected using enhanced chemiluminescence (Santa Cruz Biotechnology, Dallas, TX, USA). The β-actin expression was used as a loading control for all immunoblotting and data were expressed as arbitrary units.

### 4.9. Statistical Analysis

All data are presented as mean ± standard error of the mean (SEM). Results were compared using Student’s *t* test or Mann–Whitney test with Welch’s correction using Graph Pad Prism software (Graph Pad Software, San Diego, CA, USA). In all instances, differences were considered statistically significant when *p* < 0.05.

## 5. Conclusions

In conclusion, our study is the first to demonstrate that treatment with DMF effectively ameliorates DEP-induced lung injury, inflammation, and oxidative and nitrosative stress. Since air pollution poses a common and serious health threat worldwide, the identification of DMF as a potent antioxidant and anti-inflammatory agent in the lungs could benefit the development of therapeutic approaches against airway pollution-associated lung injury and subsequent decline in lung function. Air pollution exposure has been associated with exacerbations of chronic respiratory disorders, such as asthma and COPD [9,10]. DMF may lead to new therapies to diminish such exacerbations.

It is important to note that for an ultimate reduction or even avoidance of air pollution-related adverse health effects, the first step should be the reduction in air pollutant emissions. One important agreement that had the aim to reduce the pollutant emission was the Kyoto Protocol which lasted until 2020. For the moment, it is crucial to have new, realistic strategies and commitment of the world leaders in favor of air pollution reduction.

## Figures and Tables

**Figure 1 ijms-21-09658-f001:**
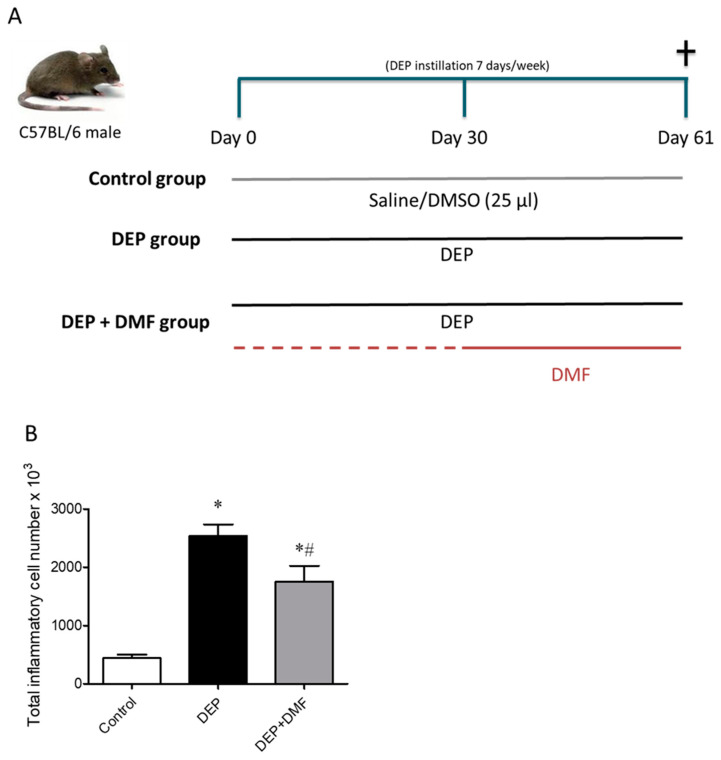
Experimental model and total inflammatory cell number in bronchoalveolar lavage (BAL). (**A**) Schematic representation of the experimental model used in our studies. Mice were either exposed to vehicle (saline/DMSO) or diesel exhaust particles (DEPs) for 60 consecutive days, with or without dimethyl fumarate (DMF) treatment for the last 30 days. (**B**) Total cell number in the bronchoalveolar lavage (BAL) of control and DEP-exposed mice, with or without DMF treatment. Data (*n* = 8) are expressed as the mean ± SEM. * *p* < 0.05 vs. control group, # *p* < 0.05 vs. DEP group.

**Figure 2 ijms-21-09658-f002:**
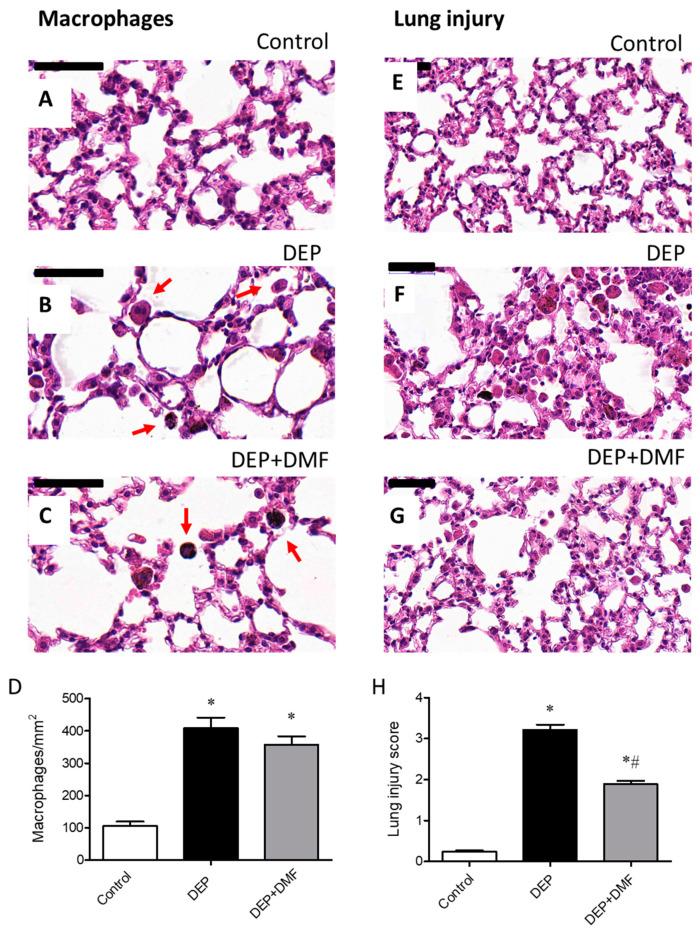
Histological analysis of the inflammatory response (macrophages) and lung injury in control and DEP-exposed mice treated with or without DMF. Representative images of macrophage distribution in mouse lungs in the control (**A**), DEP (**B**), and DEP + DMF groups (**C**). Data on the number of macrophages per mm^2^ were summarized for all 3 groups (**D**). Representative images of murine lungs in the control (**E**), DEP (**F**), and DEP + DMF (**G**) groups. Based on these images, the severity of lung injury was assessed using a 5-point scoring system (**H**). Sections were stained with hematoxylin–eosin and the bar is equal to 50 μm. Red arrows indicate macrophages in the lung sections. Data (*n* = 8) are expressed as the mean ± SEM. * *p* < 0.05 vs. control group, # *p* < 0.05 vs. DEP group.

**Figure 3 ijms-21-09658-f003:**
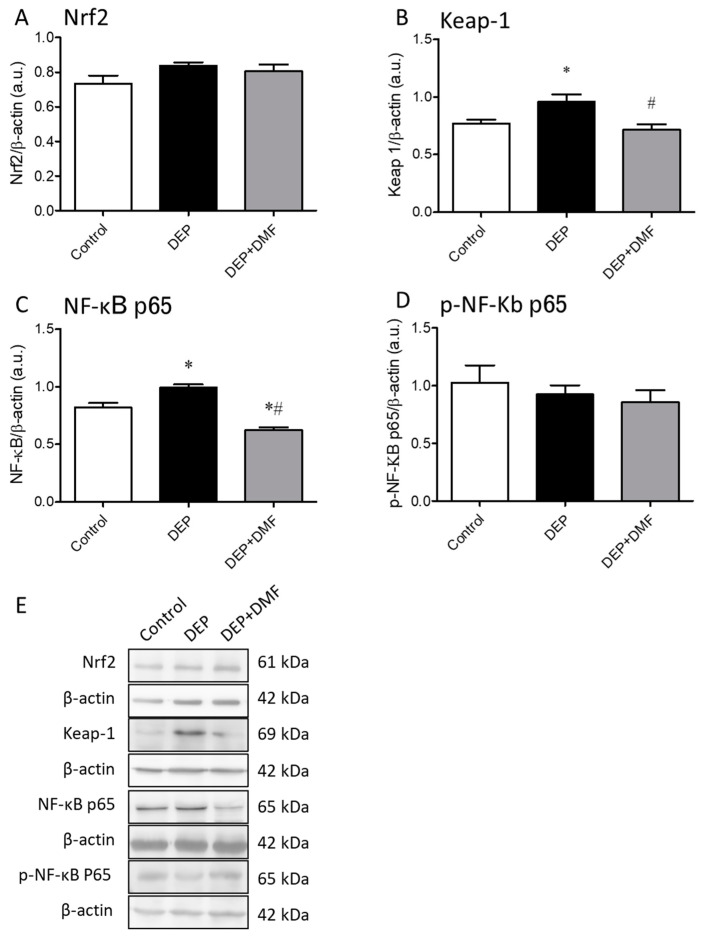
Densitometric comparison of nuclear factor erythroid 2-related factor 2 (Nrf2, (**A**)), kelch-like ECH-associated protein 1 (Keap-1, (**B**)), total nuclear factor kappa B p65 (NF-κB, (**C**)), phospho-nuclear factor kappa B p65 (p-NF-κB p65, (**D**)) protein expression in mouse lungs after exposure to DEP for 60 consecutive days, with or without DMF treatment for the last 30 days. Representative immunoblots are shown in panel (**E**). The densitometry is expressed as arbitrary units (a.u.) and β-actin was used as a loading control for all immunoblots. Except for Keap-1 (*n* = 6), data (*n* = 3) are expressed as the mean ± SEM. * *p* < 0.05 vs. control group, # *p* < 0.05 vs. DEP group.

**Figure 4 ijms-21-09658-f004:**
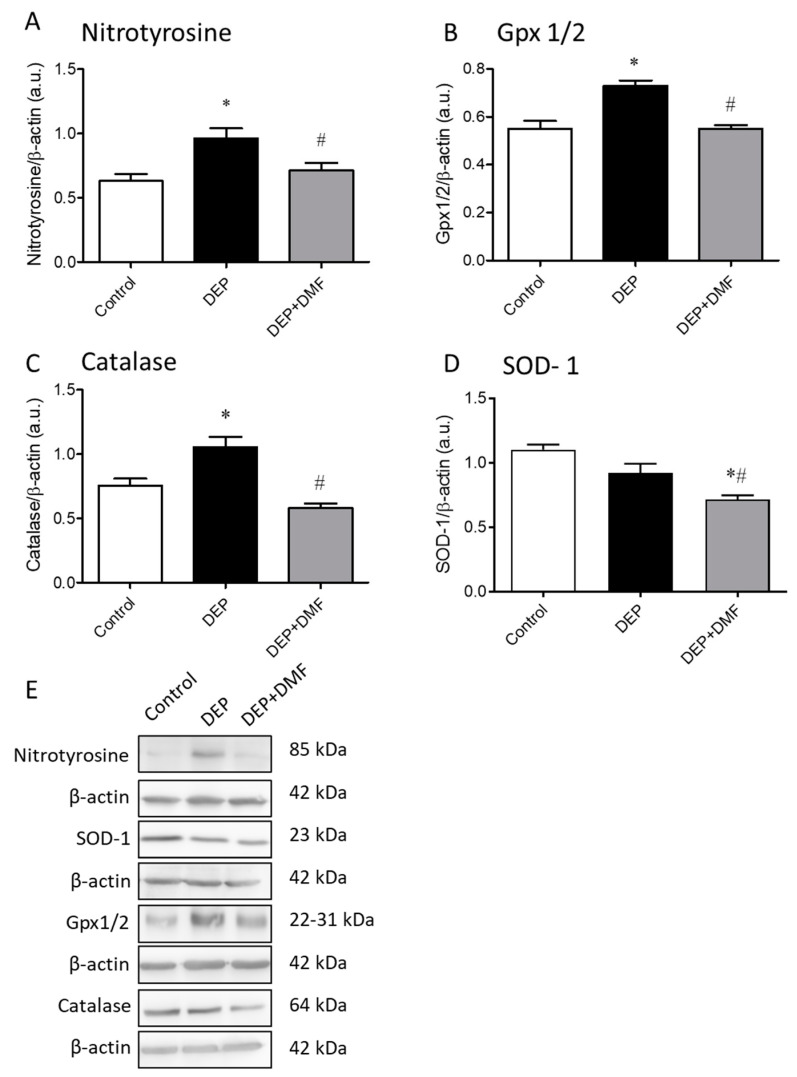
Densitometric comparison of Western blots for nitrotyrosine (**A**), glutathione peroxidase-1/2 (Gpx-1/2, (**B**)), catalase (**C**), and superoxide dismutase-1 (SOD-1, (**D**)) in mouse lungs after 60 consecutive days of exposure to DEP, with or without DMF treatment for the last 30 days. Representative immunoblots are shown in panel (**E**). The densitometry is expressed as arbitrary units (a.u.) and β-actin was used as a loading control for all immunoblots. Except for nitrotyrosine (*n* = 6), data (*n* = 3) are expressed as the mean ± SEM. * *p* < 0.05 vs. control group, # *p* < 0.05 vs. DEP group.

**Table 1 ijms-21-09658-t001:** The effects of DMF treatment on oxidative and nitrosative stress markers in BAL after chronic DEP instillation.

Parameters	Groups		
	Control	DEP	DEP + DMF
Total ROS (µg formazan/10^3^ cell)	44.81 ± 11.22	771.5 ± 111.1 *	407.9 ± 60.62 *^,#^
ROS (H_2_DCFDA) (% to control)	1.00 ± 0.22	3.02 ± 0.59 *	1.44 ± 0.27 ^#^
NO (DAF) (% to control)	1.00 ± 0.32	2.48 ± 0.40 *	1.21 ± 0.25 ^#^
OONO (APF) (% to control)	1.00 ± 0.22	3.00 ± 0.73 *	1.31 ± 0.26 ^#^

Marker assessment in BAL cells was performed as follows: total reactive oxygen species (ROS) by nitroblue tetrazolium chloride (NBT); intracellular total ROS with fluorescent probe 2′,7′-dichlorodihydrofluorescein diacetate (H_2_DCFDA); nitric oxide (NO) with fluorescent probe 4-amino-5-methylamino-2′,7′-difluorofluorescein diacetate (DAF); peroxynitrite (OONO) with fluorescent probe aminophenyl fluorescein (APF). Data (*n* = 3) are expressed as mean ± SEM. * *p* < 0.05 vs. control group, ^#^
*p* < 0.05 vs. DEP group.

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
