# Peer review of "Dimethyl Fumarate Attenuates Lung Inflammation and Oxidative Stress Induced by Chronic Exposure to Diesel Exhaust Particles in Mice"

_ijms, 2020, doi:10.3390/ijms21249658_

Round 1
Reviewer 1 Report
The manuscript “Dimethyl fumarate attenuates lung inflammation and oxidative stress induced by chronic exposure to diesel exhaust particles in mice” by Cattani-Calvieri et al. describes a study that examined the potential therapeutic effect of dimethyl fumarate (DMF) on diesel PM-induced lung damages in mice.
This is an interesting report, however the manuscript suffered from some defects as following:
Authors must provide more detailed description in the material and methods section: How many mice per group were used? How were they assigned in the groups? In addition, n=8 (8 mice?) was shown for some assessments and n=3 for other. How to decide it?
A group control “DMF” alone is missing
Please provide ref how the dose of 30 µg of DEP was chosen.
The units in the table 1 are missing.
In the legend of the figure 1, “dimethyl sulfoxide” is to correct.
The relevance of evaluating Nrf2 protein expression in whole lung lysates should be discussed. In the same way, the relevance to use DEP from the NIST, compared to PM from the current vehicle fleet should be discussed, as well as the use of a therapeutic strategy to address the effects of air pollution.
A conclusion is missing
Author Response
From:
Martina Schmidt, PhD
Department of Molecular Pharmacology
University of Groningen
Antonius Deusinglaan 1
9713 AV Groningen, The Netherlands
E-mail: m.schmidt@rug.nl
To:
Gracie Zhang
Assistant Editor, MDPI AG
Email: gracie.zhang@mdpi.com Groningen November 30, 2020
[IJMS] Manuscript ID: ijms-978495 - Major Revisions
Dear Gracie Zhang:
Please find enclosed the revision of our manuscript, entitled " Dimethyl fumarate attenuates lung inflammation and oxidative stress induced by chronic exposure to diesel exhaust particles in mice”, by Isabella Cattani-Cavalieri et al. (IJMS-978495).
A detailed description of the changes made in our revised manuscript is given in the point-to-point response below. Importantly, we have performed the requested adaptations and corrections to further increase the readability of our current manuscript. Of particular interest, we added experimental findings on the effect of DMF on DEP-induced expression of nuclear factor κB (NF-κB) p65 in its phosphorylated state. We also analyzed NF-κB p50 (total p50 and p-50). Where possible we added the additional findings. We have included a new paragraph in the discussion of our revised manuscript. Here we highlight potential limitations of our current study design, and our current research findings. However, and most important we emphasize that DMF has not been studied in experimental models exposed to air pollution - including DEP - yet. Our research findings implicate that DMF may lead to novel therapies to diminish lung injury caused by air pollutants.
We feel that we have been able to address all remarks adequately and hope that our revised manuscript is now acceptable for publication in the International Journal of Molecular Sciences, section: “Molecular Pathology, Diagnostics, and Therapeutics”. We are looking forward to your response.
With kind regards on behalf of the co-authors,
Martina Schmidt
Open Review_Review 1
Comments and Suggestions for Authors:
The manuscript “Dimethyl fumarate attenuates lung inflammation and oxidative stress induced by chronic exposure to diesel exhaust particles in mice” by Cattani-Cavalieri et al. describes a study that examined the potential therapeutic effect of dimethyl fumarate (DMF) on diesel PM-induced lung damages in mice.
This is an interesting report, however the manuscript suffered from some defects as following:
Response (R): We would like to thank the reviewers for the valuable comments. We have answered each point in the point-by-point response given below, and highlighted according changes in the manuscript.
Comment 1 (C1): Authors must provide more detailed description in the material and methods section: How many mice per group were used? How were they assigned in the groups? In addition, n=8 (8 mice?) was shown for some assessments and n=3 for other. How to decide it?
Response 1 (R1): We thank the reviewer for this valuable comment. As stated in the Material and methods section (line 331-341, page 10), we used 8 animals per group for the experiment and the animals were assigned randomly in the groups as described in detail in the following sentence, in Materials and methods section, line 331: “Animals were randomly allocated to three groups (8 animals per group) …”. However, for the measurements of ROS, NO and OONO we used 3 samples from each group, for this reason, we have n=3 for these analyses. See the following sentence added in Materials and methods section, line 357: “BAL cells from 3 randomly chosen samples of each experimental group were used for detection of...”
Additionally, we also included the information in Materials and methods section, line 376 as shown in the following sentence: “Lung lysate proteins (30 μg) from 3 randomly chosen samples of each experimental group were separated by sodium dodecylsulfate-polyacrylamide…”. As well we corrected the legend of Figure 3 of the western blots analyses, line 162: “Data (n=3) are expressed as the mean ...” And legend of Figure 4 of the western blots analyses, line 179: “Data (n=3) are expressed as the mean …”
C2: A group control “DMF” alone is missing.
R2: We thank the reviewer for his/her comment on the inclusion of a DMF group. As a matter of fact we discussed particularly this item intensively when we designed the current experimental outline. The inclusion of control groups are generally based on the points outlined below. For new drugs where signaling mechanisms are unclear or when pharmacokinetics/pharmacodynamics are unknown, we fully agree with the reviewers request to include a control group + treatment; indeed it would be mandatory. However, we think that this is not the case for dimethyl fumarate (DMF), a FDA approved drug in USA and available commercially also in other countries. Currently, about 1,416 manuscripts can be allocated in PubMed dealing with DMF (https://pubmed.ncbi.nlm.nih.gov/?term=dimethyl+fumarate&sort=date), about 11 of the total manuscripts report on studies specifically in the mouse lung (https://pubmed.ncbi.nlm.nih.gov/?term=dimethyl+fumarate+and+mouse+lung&sort=date). Of interest to note that none of the studies in mice included a control group + treatment. We think the rational here fore is as outlined. As stated DMF is already a very well-documented drug. We included details in our current manuscript, for example in the introduction, line 76 onwards, page 2 and the discussion, line 285 onwards, page 9. Reports using DMF in control group showed no toxic effects and can be found in the following manuscripts:
- Campolo, M.; Casili, G.; Biundo, F.; Crupi, R.; Cordaro, M.; Cuzzocrea, S.; Esposito, E. The Neuroprotective Effect of Dimethyl Fumarate in an MPTP-Mouse Model of Parkinson's Disease: Involvement of Reactive Oxygen Species/Nuclear Factor-kappaB/Nuclear Transcription Factor Related to NF-E2. Antioxid Redox Signal 2017, 27, 453-471, doi:10.1089/ars.2016.6800.
- Cordaro, M.; Casili, G.; Paterniti, I.; Cuzzocrea, S.; Esposito, E. Fumaric Acid Esters Attenuate Secondary Degeneration after Spinal Cord Injury. J Neurotrauma 2017, 34, 3027-3040, doi:10.1089/neu.2016.4678.
- Rojo, A.I.; Pajares, M.; Garcia-Yague, A.J.; Buendia, I.; Van Leuven, F.; Yamamoto, M.; Lopez, M.G.; Cuadrado, A. Deficiency in the transcription factor NRF2 worsens inflammatory parameters in a mouse model with combined tauopathy and amyloidopathy. Redox Biol 2018, 18, 173-180, doi:10.1016/j.redox.2018.07.006. We also emphasized now a study performed in a acetaminophen-induced hepatic injury model in mice in which DMF showed no toxic effect on liver function (for details see discussion, line 285 onwards, page 9).
Importantly, however, there are zero manuscripts about DMF and air pollution (https://pubmed.ncbi.nlm.nih.gov/?term=dimethyl+fumarate+and+air+pollution&sort=date). Based on the details outlined above we can conclude that our current study is very original and deserve publication to gain accessibility for a broader research community working on lung injury models induced by air pollutants. We have now emphasized that studies are not yet available using DMF in experimental models exposed to air pollutants. For details we would like to refer to the introduction, line 84 onwards, page 2, and the discussion, line 185 onwards, pages 7-10.
Beyond that, we have some rules for animal handle in our institution. All experimental procedures were performed in accordance to protocol reviewed and approved by the Federal University of Rio de Janeiro Institutional Animal Care and Use Committee. They put a limit number of mice that our group can purchase and this point sometimes affect our investigation. Please take in mind the three rules for animal use: reduce refine and reuse. Since the experimental design was repeated some times to ensure the quality and novelty, we had a very difficult decision to exclude some groups, for example, control + DMF and other doses of DEP and/or DMF. Finally, we found only 14 chronic studies by using rodents exposed to DEP:
- Mauderly JL, Snipes MB, Barr EB, Belinsky SA, Bond JA, Brooks AL, Chang IY, Cheng YS, Gillett NA, Griffith WC, et al. Pulmonary toxicity of inhaled diesel exhaust and carbon black in chronically exposed rats. Part I: Neoplastic and nonneoplastic lung lesions. Res Rep Health Eff Inst. 1994 Oct;(68 Pt 1):1-75; discussion 77-97. PMID: 7530965.
- Heinrich U, Muhle H, Takenaka S, Ernst H, Fuhst R, Mohr U, Pott F, Stöber W. Chronic effects on the respiratory tract of hamsters, mice and rats after long-term inhalation of high concentrations of filtered and unfiltered diesel engine emissions. J Appl Toxicol. 1986 Dec;6(6):383-95. doi: 10.1002/jat.2550060602. PMID: 2433325.
- Lim HB, Ichinose T, Miyabara Y, et al. Involvement of superoxide and nitric oxide on airway inflammation and hyperresponsiveness induced by diesel exhaust particles in mice. Free Radical Biology & Medicine. 1998 Oct;25(6):635-644. DOI: 10.1016/s0891-5849(98)00073-2.
- Sydbom A, Blomberg A, Parnia S, Stenfors N, Sandström T, Dahlén SE. Health effects of diesel exhaust emissions. Eur Respir J. 2001 Apr;17(4):733-46. doi: 10.1183/09031936.01.17407330. PMID: 11401072.
- Saito Y, Azuma A, Kudo S, Takizawa H, Sugawara I. Long-term inhalation of diesel exhaust affects cytokine expression in murine lung tissues: comparison between low- and high-dose diesel exhaust exposure. Exp Lung Res. 2002 Sep;28(6):493-506. doi: 10.1080/01902140290096764. PMID: 12217215.
- Hiramatsu K, Azuma A, Kudoh S, Desaki M, Takizawa H, Sugawara I. Inhalation of diesel exhaust for three months affects major cytokine expression and induces bronchus-associated lymphoid tissue formation in murine lungs. Exp Lung Res. 2003 Dec;29(8):607-22. doi: 10.1080/01902140390240140. PMID: 14594659.
- Kafoury RM, Madden MC. Diesel exhaust particles induce the over expression of tumor necrosis factor-alpha (TNF-alpha) gene in alveolar macrophages and failed to induce apoptosis through activation of nuclear factor-kappaB (NF-kappaB). Int J Environ Res Public Health. 2005 Apr;2(1):107-13. doi: 10.3390/ijerph2005010107. PMID: 16705808; PMCID: PMC3814704.
- Mauad T, Rivero DH, de Oliveira RC, Lichtenfels AJ, Guimarães ET, de Andre PA, Kasahara DI, Bueno HM, Saldiva PH. Chronic exposure to ambient levels of urban particles affects mouse lung development. Am J Respir Crit Care Med. 2008 Oct 1;178(7):721-8. doi: 10.1164/rccm.200803-436OC. Epub 2008 Jul 2. PMID: 18596224; PMCID: PMC2556454.
- Kim BG, Lee PH, Lee SH, Kim YE, Shin MY, Kang Y, Bae SH, Kim MJ, Rhim T, Park CS, Jang AS. Long-Term Effects of Diesel Exhaust Particles on Airway Inflammation and Remodeling in a Mouse Model. Allergy Asthma Immunol Res. 2016 May;8(3):246-56. doi: 10.4168/aair.2016.8.3.246. PMID: 26922935; PMCID: PMC4773213.
- Tyler CR, Zychowski KE, Sanchez BN, Rivero V, Lucas S, Herbert G, Liu J, Irshad H, McDonald JD, Bleske BE, Campen MJ. Surface area-dependence of gas-particle interactions influences pulmonary and neuroinflammatory outcomes. Part Fibre Toxicol. 2016 Dec 1;13(1):64. doi: 10.1186/s12989-016-0177-x. PMID: 27906023; PMCID: PMC5131556.
- Yoshizaki K, Brito JM, Moriya HT, Toledo AC, Ferzilan S, Ligeiro de Oliveira AP, Machado ID, Farsky SH, Silva LF, Martins MA, Saldiva PH, Mauad T, Macchione M. Chronic exposure of diesel exhaust particles induces alveolar enlargement in mice. Respir Res. 2015 Feb 7;16(1):18. doi: 10.1186/s12931-015-0172-z. PMID: 25848680; PMCID: PMC4345004.
- Vieira RP, Toledo AC, Silva LB, Almeida FM, Damaceno-Rodrigues NR, Caldini EG, Santos AB, Rivero DH, Hizume DC, Lopes FD, Olivo CR, Castro-Faria-Neto HC, Martins MA, Saldiva PH, Dolhnikoff M. Anti-inflammatory effects of aerobic exercise in mice exposed to air pollution. Med Sci Sports Exerc. 2012 Jul;44(7):1227-34. doi: 10.1249/MSS.0b013e31824b2877. PMID: 22297803.
- Yoshizaki K, Brito JM, Toledo AC, Nakagawa NK, Piccin VS, Junqueira MS, Negri EM, Carvalho AL, Oliveira AP, Lima WT, Saldiva PH, Mauad T, Macchione M. Subchronic effects of nasally instilled diesel exhaust particulates on the nasal and airway epithelia in mice. Inhal Toxicol. 2010 Jun;22(7):610-7. doi: 10.3109/08958371003621633. PMID: 20429853.
- Lopes FD, Pinto TS, Arantes-Costa FM, Moriya HT, Biselli PJ, Ferraz LF, Lichtenfels AJ, Saldiva PH, Mauad T, Martins MA. Exposure to ambient levels of particles emitted by traffic worsens emphysema in mice. Environ Res. 2009 Jul;109(5):544-51. doi: 10.1016/j.envres.2009.03.002. Epub 2009 Apr 10. PMID: 19362299.
This reflect the difficulty to conduct a chronic model of DEP exposure. To further increase the readability of our manuscript we added a section into the discussion to highlight potential limitations of our current study Importantly, however, we emphasized here also the novelty of our findings.
See the following details in Discussion section, page 9-10, line 270 onwards: “DMF is a well-known drug [26,27]. However, despite its well documented anti-inflammatory and antioxidant properties, DMF has not been studied in experimental models of lung injury caused by air pollutants yet. In our current we omitted the inclusion of a DMF group alone. The lack of a DMF group may represent a potential limitation of our current study. However, it is important to note that DMF administered alone to a acetaminophen-induced hepatic injury model in mice showed no toxic effect on liver function measured by biochemical markers such as serum glutamic oxaloacetic transaminase, serum glutamic pyruvic transaminase, gamma-glutamyl transferase, alkaline phosphatase, bilirubin and albumin in serum [28], implicating that DMF alone does not exhibit unwanted side effects. However, and most important, we administered DMF by oral gavage in chronically DEP exposed mice to more closely resemble clinical procedures. Upon addition of DMF after 30 days of DEP exposure we probably mimic a ‘real life’ situation more closely. We are constantly exposed to air pollution, rendering pretreatment impossible, and typically only seek help after worsening of disease symptoms such as exacerbations of asthma and COPD patients [4,7,8]. It is important to note that DMF reduced lung inflammation and oxidative and nitrosative stress induced by chronic DEP exposure, processes known to be difficult to control and contribute to decline in lung function in asthma and COPD patients. It is also important to note that we used in our current study DEP from NIST instead particulate matter from buses [25], to increase the probability to reproduce our current research findings by research groups world-wide.”
C3: Please provide ref how the dose of 30 µg of DEP was chosen.
R3: We thank the reviewer for this valuable comment. In order to have a relevant model, trying to mimic the “real” life, we choose chronic model with a low dose of 30 µg of DEP for intranasal instillation as shown in the following sentence in discussion section, line 163: “An important finding of our work is that exposure of mice to a relatively low dose (30 µg) of DEP [23] (without any additional stimulus) for 60 consecutive days was able to promote lung injury.” We added the referred article in discussion section, line 163: [23] Yoshizaki, K., Brito, J.M., Moriya, H.T. et al. Chronic exposure of diesel exhaust particles induces alveolar enlargement in mice. Respir Res 16, 18 (2015). https://doi.org/10.1186/s12931-015-0172-z
C4: The units in the table 1 are missing.
R4: We thank the reviewer to point out this mistake. We have adopted the table accordingly in Result section, page 5, line 127.
C5: In the legend of the figure 1, “dimethyl sulfoxide” is to correct.
R5: We thank the reviewer to point out this mistake. The manuscript has been adopted accordingly.
C6: The relevance of evaluating Nrf2 protein expression in whole lung lysates should be discussed. In the same way, the relevance to use DEP from the NIST, compared to PM from the current vehicle fleet should be discussed, as well as the use of a therapeutic strategy to address the effects of air pollution.
R6: We thank the reviewer for this valuable comment. We agree that the evaluation of Nrf2 protein expression might envisioned as a potential limitation of our study. Studies in whole lung lysates have disadvantages; mainly because we are processing different types of cells, such as type I and type II cells, endothelial cells, fibroblasts and of course immune cells. However, all cell types in the lung can express Nrf2. We are not claiming that Nrf2 is from one or other cell type specifically, but we are reporting the findings as a “whole lung” expression based on DEP stimulation. Since DMF is an Nrf2 inducer factor, we are assuming that this drug can affect all cells. Pay attention regarding whole lung lysates this is a common procedure for lung analyses and was addressed here: Micromachines (Basel). 2017 Mar; 8(3): 83. doi: 10.3390/mi8030083. We referred throughout our manuscript to the “whole lung lysate”, for details see for example in Result section, page 5, line 136 onwards: “To evaluate the effects of chronic DEP exposure and DMF treatment on the Nrf2 pathway, we measured Nrf2 and Keap-1 protein expression in whole lung lysates by western blotting.”
We used DEP from NIST in this study based on the rational that other research groups around the world have access to the same compound. Previously we published a manuscript with PM obtained from buses (Cattani-Cavalieri, I.; Valenca, S.S.; Lanzetti, M.; Carvalho, G.M.C.; Zin, W.A.; Monte-Alto-Costa, A.; Porto, L.C.; Romana-Souza, B. Acute Exposure to Diesel-Biodiesel Particulate Matter Promotes Murine Lung Oxidative Stress by Nrf2/HO-1 and Inflammation Through the NF-kB/TNF-alpha Pathways. Inflammation 2019, 42, 526-537, doi:10.1007/s10753-018-0910-8.), but the effect observed at that time in mouse lung is maybe unique since each city around the world have different access to different diesel resources. Engine, maintenance, weather and humidity - all variables that make our result unique with little chance to compare with other studies. By using DEP from NIST we will have a chance to compare our results with other manuscripts and also to make our experimental design reproducible by other researches groups. We added in our revised manuscript details about the different air pollutants used in different studies. For details please refer to Introduction section, line 47 onwards, page 2.
Our current study implicate that DMF may lead to novel therapies to prevent exacerbations in asthma and COPD. We have emphasized this aspect in the final section of the discussion, for details see line. In addition we added now detail in the conclusion section. For example. See details in Discussion section, page 9-10, line 272 onwards. In addition we have rephrased the conclusion of our revised manuscript, for details we refer to page 12, line 372 onwards.
C7: A conclusion is missing
R7: We thank the reviewer to point out this mistake. We corrected the formatting problem, adding the conclusion in the conclusion section. We have included more details into the current conclusion section. For details we refer to the Conclusion section, page 12, line 372 onwards.
Reviewer 2 Report
- line 19 abstract: 30 Ug/mice should be per mouse
- The impact DMF on macrophage infiltration needs further explanation, to help the reader understand the lack of effect.
- Figure 2 : the figures orientation and labelling is a bit confusing, horizontal view for a,b,c and then for d,e,f with the proper title above each group might be better.
- Table 1: the units of quantification needs to be added
- Table 1 : in the legends, DCF, DAF and APF needs to be spelled out
- line 112 : What is the explanation for this effect, below baseline level of NFKB, this needs to be further explained, any attempts to measure and quantify the phosphorylated NFKB
- The quality of the western blots is not acceptable, the housekeeping protein seems to be overexposed in figure 3 and 4
Author Response
From:
Martina Schmidt, PhD
Department of Molecular Pharmacology
University of Groningen
Antonius Deusinglaan 1
9713 AV Groningen, The Netherlands
E-mail: m.schmidt@rug.nl
To:
Gracie Zhang
Assistant Editor, MDPI AG
Email: gracie.zhang@mdpi.com Groningen November 30, 2020
[IJMS] Manuscript ID: ijms-978495 - Major Revisions
Dear Gracie Zhang:
Please find enclosed the revision of our manuscript, entitled " Dimethyl fumarate attenuates lung inflammation and oxidative stress induced by chronic exposure to diesel exhaust particles in mice”, by Isabella Cattani-Cavalieri et al. (IJMS-978495).
A detailed description of the changes made in our revised manuscript is given in the point-to-point response below. Importantly, we have performed the requested adaptations and corrections to further increase the readability of our current manuscript. Of particular interest, we added experimental findings on the effect of DMF on DEP-induced expression of nuclear factor κB (NF-κB) p65 in its phosphorylated state. We also analyzed NF-κB p50 (total p50 and p-50). Where possible we added the additional findings. We have included a new paragraph in the discussion of our revised manuscript. Here we highlight potential limitations of our current study design, and our current research findings. However, and most important we emphasize that DMF has not been studied in experimental models exposed to air pollution - including DEP - yet. Our research findings implicate that DMF may lead to novel therapies to diminish lung injury caused by air pollutants.
We feel that we have been able to address all remarks adequately and hope that our revised manuscript is now acceptable for publication in the International Journal of Molecular Sciences, section: “Molecular Pathology, Diagnostics, and Therapeutics”. We are looking forward to your response.
With kind regards on behalf of the co-authors,
Martina Schmidt

Round 2
Reviewer 1 Report
The reviewer thanks the authors for their answers and the improvement of their manuscrit. However, I have still some remarks/interrogations:
- The DEP from the NIST have been well studied, but are not representative of the current fleet, due to the evolutions of regulations and consequently, of the motor technologies. It is a limitation of this study that needs to be discussed.
- The conclusion is too speculative. The authors claimed that this study “is of particular interest in the context of acute worsening of asthma…”. The study was conducted from healthy mice. The treatment with DMF was started 30 days after the first instillation of DEP. Is this experimental design based on the development of airway diseases, such as asthma at day 30? There is also the issue of “airway pollution associated malignancies”. The aim must be clarify.
- The first line of action to avoid or limited pollution-related diseases is to reduce emissions. It is an important message to keep in mind and which could be added in the conclusion.
- It is odd to perform some evaluations with 3 randomly chosen samples, particularly for the fluorescent studies.
Author Response
From:
Martina Schmidt, PhD
Department of Molecular Pharmacology
University of Groningen
Antonius Deusinglaan 1
9713 AV Groningen, The Netherlands
E-mail: m.schmidt@rug.nl
To:
Gracie Zhang
Assistant Editor, MDPI AG
Email: gracie.zhang@mdpi.com Groningen December 13, 2020
[IJMS] Manuscript ID: ijms-978495 - Major Revisions
Dear Gracie Zhang:
Please find enclosed the revision of our manuscript, entitled "Dimethyl fumarate attenuates lung inflammation and oxidative stress induced by chronic exposure to diesel exhaust particles in mice”, by Isabella Cattani-Cavalieri et al. (IJMS-978495).
A detailed description of the changes made in our revised manuscript is given in the point-to-point response below. Importantly, we have performed additional experiments to further substantiate our experimental findings on the level of the protein expression of Keap-1 and nitrotyrosine in whole lung lysates, and included the new analysis into the revised Figure 3 and Figure 4. We have included into the discussion now a statement about the current evolution of motor technologies in developed and developing countries. We have now re-written our conclusion to avoid too speculative features of the outlook.
We feel that we have been able to address all remarks adequately and hope that our revised manuscript is now acceptable for publication in the International Journal of Molecular Sciences, section: “Molecular Pathology, Diagnostics, and Therapeutics”. We are looking forward to your response.
With kind regards on behalf of the co-authors,
Martina Schmidt

Reviewer 2 Report
The authors addressed all comments, line 19 I still think this should be dose per mouse
Author Response

(The authors gave the same response as above.)

Round 3
Reviewer 1 Report
In my opinion, this manuscript has been well improved and is now suitable for publication.